# Fabrication and Study of Dextran/Sulfonated Polysulfone Blend Membranes for Low-Density Lipoprotein Adsorption

**DOI:** 10.3390/ma16134641

**Published:** 2023-06-27

**Authors:** Fei Fang, Hai-Yang Zhao, Rui Wang, Qi Chen, Qiong-Yan Wang, Qing-Hua Zhang

**Affiliations:** 1College of Chemical and Biological Engineering, Zhejiang University, Hangzhou 310027, China; 3090102659@zju.edu.cn; 2Research and Development Center, Zhejiang Sucon Silicone Co., Ltd., Shaoxing 312088, China; hycsea@163.com (H.-Y.Z.); 13989503732@163.com (R.W.); 13989552135@163.com (Q.C.)

**Keywords:** atherosclerosis, polysulfone, low-density lipoprotein, enzyme-linked immunosorbent assay

## Abstract

The abnormal increase in low-density lipoprotein (LDL) in human blood is a main independent risk factor for the pathogenesis of atherosclerosis, whereas a reduced LDL level effectively lowers morbidity. It is important to develop LDL adsorption materials with high efficiency and selectivity, as well as to simplify their fabrication processes. In this paper, polysulfone (PSF), sulfonated polysulfone (SPSF), and sulfonated polysulfone/dextran (SPSF/GLU) membranes were successfully fabricated for LDL adsorption using a solution casting technique. Attenuated total reflectance Fourier transform infrared spectroscopy and X-ray photoelectron spectroscopy measurements confirmed the success of the preparation. The water contact angle decreased from 89.7 ± 3.4° (PSF) to 76.4 ± 3.2° (SPSF) and to 71.2 ± 1.9° (SPSF/GLU), respectively. BSA adsorption testing showed that the SPSF/GLU with surface enrichment of sulfonate groups and glycosyl groups possessed higher resistance to protein solution. The adsorption and desorption behaviors of the studied samples in single-protein or binary-protein solutions were systematically investigated by enzyme-linked immunosorbent assay (ELISA), The results showed that SPSF/GLU, which had excellent resistance to protein adsorption, possessed a similar adsorption capacity to that of PSF. SPSF membrane exhibited excellent selective affinity for LDL in single and binary protein solutions, suggesting potential applications in LDL removal.

## 1. Introduction

Coronary artery disease (CAD), the leading cause of mortality worldwide, primarily arises from atherosclerosis [1,2]. Although the underlying pathogenesis of atherosclerosis is not yet fully understood, abnormal levels of low-density lipoprotein (LDL) in the bloodstream have consistently been regarded as the primary pathogenic factor contributing to the progression of atherosclerosis [3,4], while some other factors (oxidized LDL (oxLDL), lipoprotein(a) (LP(a)), apolipoprotein E4, etc.) have also been reported [5,6,7]. Currently, lifestyle changes [8,9,10] and drug therapy (e.g., statins [11], bempedoic acid [12], inclisiran [13]) are the two most prevalent clinical interventions used to regulate the LDL levels. However, due to their side effects, long-term use of drugs often leads to problems such as acute coronary syndrome and muscle weakness [14,15]. Moreover, these aforementioned therapies are ineffective for patients who are resistant to drugs or suffering from severe hyperlipidemia or familial hypercholesterolemia [16,17].

LDL apheresis, based on extracorporeal blood purification techniques, is highly recommended under such circumstances for its high efficiency in reducing LDL cholesterol and meeting clinical end points [18,19]. Currently, there are six common LDL apheresis technologies available, four of which are based on the principle of adsorption: immune-adsorption (IMA), dextran sulfate adsorption (DSA), polyacrylate-coated polyacrylamide direct perfusion (DALI), and dextran sulfate direct perfusion (Liposorber D) [20,21,22]. The efficacy of LDL apheresis systems is primarily determined by the adsorbents. Therefore, extensive attention has been drawn to the investigation of economical and effective adsorbents, leading to the development of various LDL ligands, such as antibodies [23,24], heparin [25,26,27], chondroitin sulfate [28,29], cyclodextrin [30,31], chitosan derivatives [32,33], phospholipid [34,35], etc.

While most researchers are committed to modifying the functional ligands on adsorbent substrates to enhance the selectivity and adsorption efficiency of adsorption materials, the relevant mechanisms are seldom explored. Generally, currently available LDL adsorbents are designed to be anionic to selectively bind to positively charged regions of apolipoprotein-B100 in LDL, based on electrostatic interaction [36,37]. Inspired by the saccharide-like structures shared in most LDL adsorption materials, in our previous work, a “multiple-interaction model” for LDL in contact with adsorbents containing negative ligands and neutral saccharides was proposed by Li et al. [38]. It was found that saccharides or saccharide-like structures also play significant roles in LDL recognition. Subsequently, anionic glycosylated PSF membranes, of which surface chemistries regulated by click reactions and the selection of an appropriate glycosyl-to-sulfonyl feed, were fabricated for affinity adsorption for LDL by Fang et al. [39]. Despite its excellent adsorption properties, the complex and time-consuming preparation process severely limits its practical application. Generally speaking, the selection of ligands and substrates, as well as the preparation methods, are crucial for the development and application of efficient LDL adsorption materials. Previous studies have shown that anionic or anionic-glycosylated ligands often exhibit excellent LDL adsorption capabilities. Various strategies have been developed to achieve facile and reliable immobilization of anionic or anionic-glycosylated biomaterial surfaces. Despite their good LDL adsorption performance of these adsorbents, the preparation process for them is often cumbersome, typically involving chemical methods for substrate calcination, severely limiting the practical application of LDL materials. Solution casting is a widely used technique for producing polymer films, coatings, and membranes. It offers convenience, cost-effectiveness, and time-saving benefits, allowing for the regulation of material properties through the utilization of additives. Moreover, it can be readily adapted for both laboratory-scale research and large-scale industrial production. Polysulfone is currently the most widely applied polymer, and our research group has devoted significant efforts to using PSF membranes for improved hemodialysis with simultaneous LDL removal. In this study, we prepared anionic or anionic-glycosylated PSF membranes for LDL adsorption. Sulfonated polysulfone and dextran were used as additives, and the membranes were prepared via the solution casting method. Attenuated total reflectance Fourier transform infrared spectroscopy (ATR-FTIR) and X-ray photoelectron spectroscopy (XPS) were performed to characterize the surfaces’ structures and chemical compositions. The surface properties of the studied samples were analyzed by surface zeta potential, water contact angle (WCA), and bovine serum albumin (BSA) adsorption measurements. Enzyme-linked immunosorbent assay (ELISA) was performed to evaluate the LDL adsorption and desorption performances of membranes, and the interaction between membrane surface and LDL was investigated.

## 2. Materials and Methods

### 2.1. Materials

Polysulfone and sulfonated polysulfone granules were obtained from Zhejiang Qinyuan Water Treatment S.T. Co., Ltd. (Shanghai, China). Low density lipoprotein (LDL, 98%) was purchased from Millipore (Burlington, MA, USA). Primary antibody anti-β-lipoprotein, secondary antibody anti-chicken, IgG and human serum albumin (HSA) were provided by Sigma-Aldrich (Saint Louis, MO, USA). Dextran (Mw = 70,000) was purchased from Aladdin (Shanghai, China). All other reagents were commercially supplied by Sinopharm Chemical (Shanghai, China) and used as received. The ultrapure water (18.2 MΩ·cm) used in all experiments was freshly prepared with an ELGA Classic UF system (Veolia Water Systems, Aubervilliers, France).

### 2.2. Preparation of Dense Membranes

A solution of SPSF at a concentration of 16 wt.% in NMP was prepared by dissolving pure SPSF granules in NMP. After complete dissolution of SPSF, 3 wt.% dextran was introduced as an additive, followed by vigorous stirring at approximately 80 °C for 24 h to achieve a homogeneous SPSF/GLU solution. When all air bubbles were completely removed from the solution, the SPSF/GLU solution was cast onto a clean glass plate using a casting knife with a thickness of 150 μm. The glass plates containing the nascent film were dried directly under a vacuum (60 °C, 24 h) and then immersed in ultrapure water. After being peeled from the glass plate, the dense films were washed with distilled water several times to remove the excess NMP solvent and then were dried for another 24 h at 80 °C under vacuum a to ensure complete dryness. Dense films of PSF or SPSF were separately prepared from a 16 wt.% PSF solution or a 16 wt.% SPSF solution using the same method.

### 2.3. Characterization

#### 2.3.1. FT-IR/ATR and XPS

ATR–FTIR measurements were performed on a Nexus 470 FTIR spectrometer (Thermo Nicolet, Detroit, MI, USA) equipped with an ATR cell (ZnSe, 45°). Thirty-two scans were performed for each spectrum at a nominal resolution of 4 cm^−1^. The chemical composition of the various PSF dense films was further analyzed by X-ray photoelectron spectroscopy (XPS) measurements, which was performed on a PHI-5000C ESCA system (Perkin Elmer, Waltham, MA, USA) with Mg Kα radiation (hν = 1253.6 eV). The maximum pressure in the analysis chamber was maintained at 10^−6^ Pa. All survey and core-level spectra were referenced to the C1s hydrocarbon peak at 284.7 eV to compensate for the surface charging effect.

#### 2.3.2. WCA Measurement

The hydrophilicity of the dense films was investigated on the basis of the water contact angle using the sessile drop method. WCA was measured on a CTS-200 contact angle goniometer (MAIST Vision Inspection & Measurement Co., Ltd., Ningbo, China) at room temperature while equipped with video capture. Typically, a 2-μL droplet of ultrapure water was dropped onto a dry sample, and then the WCA was calculated from the digital image using DropMeter software (version 2.0). At least five independent results were averaged to obtain one mean value.

#### 2.3.3. Zeta Potential Measurements

The surface potential of the studied samples was determined by measuring the zeta potential using a zeta potential analyzer (Delsa™, Beckman Coulter, Brea, CA, USA). A 2 cm × 3 cm strip was randomly cut from the samples and thoroughly washed with ultrapure water. The strip was subsequently immersed in NaCl solution (0.1 M, pH 7.0) at 25 °C for 45 min and then was detected using 10^−3^ M KCl solution (pH 7.4).

#### 2.3.4. BSA Adsorption

Various concentrations of BSA (0.25 mg/mL, 0.50 mg/mL, 0.75 mg/mL, and 1.0 mg/mL) were dispersed in PBS (137 mmol/L NaCl; 1.4 mmol/L KH_2_PO_4_; 4.3 mmol/L Na_2_HPO_4_; 2.7 mmol/L KCl, adjusted to pH 7.4 with 1 M NaOH) and a standard curve was obtained for absorbance at 280 nm (Appendix A). In a typical experiment, the film was immersed in 4 mL of the BSA solution and was gently shaken for 2 h at 37 °C. The absorbance of the membrane-immersed BSA solution was measured, and the quantity of BSA adsorbed was calculated using the standard curve. Each value reported is an average of at least three independent measurements.

#### 2.3.5. ELISA for LDL Adsorption and Desorption

Adsorption of LDL on the different films was investigated by ELISA (Appendix A) as described in our previous work [25,26,27]. All solutions were freshly prepared before each measurement. LDL or human serum albumin (HSA) was dissolved in PBS. The primary antibody (anti-β-lipoprotein) and the secondary antibody (the peroxidase-conjugated secondary antibody, anti-chicken IgY (IgG)) were diluted in a 0.1 wt.% BSA solution in PBS (pH 7.4) to concentrations of 1:5000 and 1:10,000, respectively. Blocking solution was prepared using a 1 wt.% BSA solution in PBS (pH 7.4). The substrate solution was freshly prepared by adding 400 μL of TMB (0.5 wt%, dissolved in DMSO) and 2 μL of hydrogen peroxide (30%) to 10 mL substrate buffer solution (Na_2_HPO_4_ and 0.1 M citric acid, adjusted to a pH of 5.0 with 1 M NaOH). The films were cut into discs with a diameter of 1.4 cm and were placed in 24-well tissue culture plates, and then 0.5 mL of LDL solutions at specified concentrations were added. The plates were incubated for 1 h at 37 °C, followed by three washes with 1 mL of PBS (pH 7.4). Subsequently, a blocking solution (0.5 mL) was added and incubated for 0.5 h at 37 °C. After rinsing with PBS (pH 7.4) a further three times, the primary and secondary antibodies were added and incubated for 1 h at 37 °C. Each subsequent step was followed by washing three times with Tris-buffered saline (TBS, 137 mM NaCl, 1.4 mM KH_2_PO_4_, 4.3 mM Na_2_HPO_4_, 2.7 mM KCl, pH 7.4, 0.1% Tween 20). Subsequently, the films were transferred to a new 24-well plate, followed by the addition of TMB substrate solution. After 10 min, a H_2_SO_4_ solution (1 M, 0.5 mL) was added to stop the chromogenic reaction. Finally, the optical density of the dye solution was measured at 450 nm with a plate reader (ELx800 Reader, BioTek, Winooski, VT, USA).

LDL desorption experiments were conducted following the same procedure used for LDL adsorption analysis with ELISA described above. However, before the addition of primary antibodies, the LDL-incubated membrane was washed with NaCl solutions of various concentrations ranging from 0.125 M to 2 M.

## 3. Results and Discussion

### 3.1. FT-IR/ATR and XPS for Films

In this paper, ATR–FTIR measurements was used to analyze the surface layer of the studied dense films, and they have been widely used to study the chemical structures of polymer surfaces. As shown in Figure 1, the spectrum of PSF is significantly different from that of SPSF and SPSF/GLU in the fingerprint region. The presence of sulfonic acid groups on the membrane surface led to the emergence of a new absorption peak at 1027 cm^−1^ in SPSF, which can be assigned to the symmetric stretching vibration of O=S=O aroused by–SO_3_H, compared with the PSF film. As for SPSF/GLU, the surface was enriched with sulfonate groups and glycosyl groups. Correspondingly, a strong broad band from 3200 cm^−1^ to 3600 cm^−1^ corresponding to the O–H stretching vibration was observed, accompanied by the diminishment of the O=S=O stretching vibration.

The surface elemental composition of the samples was further quantified by XPS analysis. Figure 2 displays the XPS spectra of the studied samples, while Table 1 presents the results derived from the XPS survey scans. No additional peaks were observed for SPSF and SPSF/GLU compared with PSF (Figure 2), mainly because the three membranes have the same composition elements except for different element content. In Table 1, it can be seen that the sulfur content increased from 2.28% on PSF to 3.32% on SPSF and 2.71% on SPSF/GLU. This increase is undoubtedly attributable to the presence of sulfate groups on the surfaces of SPSF and SPSF/GLU. Meanwhile, the oxygen content of SPSF increased from 15.91% to 19.62% compared with that of PSF, as the sulfate groups exhibit relatively high oxygen content. As for SPSF/GLU, owing to the emergence of substantial hydroxyl groups on the surface, the oxygen content increased significantly to 21.76%, indicating that dextran had been modified on the membrane surface.

### 3.2. Hydrophilicity of the Membrane Surface

One of the most crucial characteristics of membranes is hydrophilicity, which has a significant impact on the antifouling capabilities of membranes. As shown in Figure 3, the initial WCA of the PSF membrane could reach almost 90°, consistent with other studies reported [40]. The introduction of sulfonic acid groups resulted in increased hydrophilicity of SPSF films compared to PSF films, while for SPSF/GLU films, the water contact angle of SPSF/GLU decreased further to 71.2 ± 1.9° due to the presence of highly hydrophilic dextran molecules. Since the studied dense films were prepared by the solvent casting method, the hydrophobic components in the casting solution were more susceptible to exposure to the air, whereas the hydrophilic sulfonic acid groups tended to be wrapped in the membrane during the film-forming process. Therefore, the studied samples were treated with another immersion in ultrapure water (80 °C, 24 h) to expose the hydrophilic groups to the interface. It could be seen that the WCA of PSF membrane remained unchanged, while the WCA of SPSF films decreased from 82.5 ± 1.3° to 76.4 ± 3.2° due to further exposure to hydrophilic sulfonic acid groups after treatment. In contrast, the WCA of SPSF/GLU increased obviously, indicating that the hydrophilicity of SPSF/GLU was decreased, which may due to the highly hydrophilic dextran being partly dissolved in water.

### 3.3. Surface Electric Properties of the Membrane Surface

Electrostatic interactions are believed to play an important role in LDL adsorption processes. Thus, the surface potential of the membrane surface was measured by zeta potential measurements. The surface ζ potentials of PSF, SPSF, and SPSF/GLU membranes were −10.91 mv, −48.52 mv, and −28.32 mv, respectively, as shown in Figure 4. Since the surface of SPSF membranes are enriched with a large number of electronegative sulfate groups, the SPSF films possessed the strongest surface electronegativity. When dextran was introduced to the membrane surface, the surface potential of SPSF/GLU obviously increased, indicating that the electronegativity of SPSF/GLU is reduced due to the relatively fewer sulfate groups covering the surface layer compared with that of SPSF.

### 3.4. BSA Adsorption

Biofilm development on implants and catheters can have detrimental effects on biomedical applications, leading to infections and antibiotic resistance. Despite decades of research on “protein resistant surfaces”, biofouling remains a limiting factor in the reliable performance of biomaterials. Materials with excellent anti-specific protein adsorption ability can largely prevent the occurrence of biofouling and improve their biocompatibility [41,42]. Here, BSA adsorption experiments were performed to evaluate the anti-nonspecific adsorption properties of the studied membranes. The mass of BSA adsorbed on the membrane was calculated using a standard curve (Appendix A), as described in our previous work [39]. The amounts of BSA adsorbed on different samples are demonstrated in Figure 5. The hydrophobic PSF membrane attracted the largest amount of BSA when the concentration of BSA was 1.5 mg/mL. Compared with the PSF membrane, the amount of BSA adsorbed decreased significantly, from 45.68 μg/cm^2^ for PSF to 26.54 μg/cm^2^ for SPSF and 17.72 μg/cm^2^ for SPSF/GLU. Since the isoelectric point of BSA is 4.7, when pH = 7.4, BSA is negatively charged. The surface electronegativity of SPSF and SPSF/GLU is significantly higher, resulting in stronger electrostatic repulsion from BSA. Furthermore, the improvement of hydrophilicity of SPSF and SPSF/GLU also prevented the adsorption of BSA.

### 3.5. LDL Adsorption and Desorption Analysis by ELISA

ELISA has been widely used to quantify specific proteins (e.g., albumin, fibrinogen, fibronectin, antibodies, LDL, et al.) that adsorb onto biomaterial surfaces from complex protein mixtures or plasma due to its reliability, sensitivity, and reproducibility [43,44,45,46]. Here, the LDL adsorption and desorption on membrane surface were investigated by ELISA, and the quantity of LDL adsorbed on studied samples membranes was estimated by optical densities from colorimetric ELISA [40,46]. Figure 6 shows the results of the adsorption of LDL on different membrane surfaces from a single protein solution. The PSF, SPSF, and SPSF/GLU dense films exhibited varying adsorption capacities for LDL, while for all samples, the quantity of LDL absorbed increased with increasing LDL concentrations until it reached a plateau. This typical Langmuir-type adsorption behavior was similar to that of some other proteins that adsorb on biomaterials [46,47,48]. As depicted in Figure 6, the absorbance of SPSF films was highest compared with the other two films when exposed to the same LDL solution, indicating that the SPSF films had the highest adsorption capacity for LDL. Due to the presence of sulfonyl groups, the negative charge density on the SPSF surface was significantly increased. It can be presumed that electrostatic attraction forces were primarily responsible for the intense adsorption of LDL onto SPSF films. Theoretically, SPSF/GLU films were expected to exhibit excellent adsorption performance due to the abundance of glycosyl and sulfonyl groups on their surfaces, which can engage in multiple interactions, including hydrogen bonding and electrostatic interactions with LDL [38]. However, SPSF/Glu showed a lower adsorption capacity for LDL than PSF and SPSF. This outcome can be attributed to the lower electronegativity of SPSF/Glu films compared with SPSF, as revealed by zeta potential measurements. Furthermore, SPSF/Glu films were much more hydrophilic than PSF, enabling the formation of a hydration layer that hindered LDL contact with the membrane surface. Additionally, the partially dissolved dextran as a polysaccharide (revealed by WCA), which has been reported to interact with LDL, as reported in the literature [38,39], may also hinder the adsorption of LDL from solution to the membrane surface. It should be noted that SPSF/GLU, which has good anti-nonspecific protein adsorption properties, exhibited a similar affinity for LDL as PSF. This finding indicated that SPSF/GLU could selectively remove LDL under pathological conditions.

Although the interaction between LDL and LDL adsorbents are not completely understood, electrostatic interactions are known to play an important role in LDL adsorption processes, while the hydrophobic interaction is another factor that should not be ignored [49]. Electrostatic interaction largely depends on ionic strength, whereas the hydrophobic interaction between the protein and the surface remains unaffected. Here, NaCl solutions with varying ionic strengths were introduced as eluants to disturb the electrostatic binding between the membrane and the adsorbed LDL. Figure 7 illustrates the results of LDL desorption from the studied samples. As can be seen, the absorbed LDL on nascent PSF was almost unaffected by NaCl concentrations up to 2 mol/L. Although the PSF membrane exhibited a negatively charged surface, it appeared that the effect of electrostatic interaction was limited. Compared with PSF, a notable reduction in LDL desorption from SPSF was observed as the concentration of NaCl increased. LDL desorption from SPSF/GLU was negligible because of the initial low adsorption capacity of SPSF/GLU for LDL, as Figure 6 shows. These results suggested that the adsorption of LDL on PSF film was mainly driven by hydrophobic interactions, whereas LDL adsorption on SPSF film was predominantly induced by electrostatic interaction.

Protein adsorption on biomaterial surfaces in blood or plasma is typically accompanied by protein competition. However, when complex protein mixtures are utilized to replicate such competitive processes, it is very challenging to control all of the critical variables. Therefore, adsorption studies from mixtures of two proteins are typically employed to simplify evaluation [50,51,52]. Here, human serum albumin (HSA) was selected as a competing protein because of its high concentration in plasma. Mixed solutions of LDL and HSA at a variety of concentrations were freshly prepared, and then the studied samples were submerged. According to the Vroman effect [52], there is a redistribution of adsorbed proteins. Proteins with higher mobility or higher concentrations initially dominated the surface due to their rapid diffusion rates but later were replaced by other proteins with higher affinity.

Figure 8 illustrates the competitive adsorption of binary protein solutions containing HAS and LDL on membrane surfaces. When the concentration of LDL was fixed at 10 μg/mL while varying the concentration of HSA from 0 to 10 mg/mL, the amount of adsorbed LDL significantly decreased for PSF and SPSF/GLU films due to the presence of HAS. However, a lesser effect was observed for SPSF films (Figure 8). Furthermore, the interference effect of HAS concentrations on the adsorption of LDL by SPSF gradually diminished. In particular, the SPSF films attracted a large amount of LDL, even under the interference of high concentrations of HSA (10 mg/mL), while the PSF and SPSF/Glu films showed negligible adsorption under the same conditions. It should be noted that the outcome reflects an HSA to LDL ratio of 1000:1, which is much higher than that in healthy human plasma (the levels of HSA and LDL were 30–50 g/L and 100–120 mg/mL [53,54], respectively). Then, the concentration of HSA was fixed at 1.0 mg/mL, while the concentration of LDL was changed from 0 to 10 μg/mL, as shown in Figure 9. Again, the binding of LDL to SPSF was much higher than to PSF and SPSF/GLU. The quantity of adsorbed LDL on PSF and SPSF/GLU increased slightly when compared to SPSF, which displayed a notable, sharp elevation of the adsorption curve.

Since the amounts of adsorbed proteins largely depend on the strength of attraction between the protein and the surface of the material, it was found that the specific interaction between LDL and SPSF was much stronger than the non-specific interactions between LDL and PSF surfaces. The SPSF film can selectively absorb LDL from binary protein solutions, making it a potent material for LDL removal. As for the SPSF/GLU film, the change in absorbance was not obvious due to its limited LDL adsorption. Additionally, dextran dissolved in LDL solutions was also a disturbing factor that could not be ignored.

## 4. Conclusions

In this work, PSF, SPSF, and SPSF/GLU dense membranes were successfully fabricated by the solution casting technique, followed by extensive tests, such as ATR spectroscopy, XPS spectroscopy, WCA, surface ζ potential measurement, etc., to characterize the chemical composition and properties of the membranes quantitatively and qualitatively. ATR-FTIR and XPS results demonstrated that the surfaces of resultant SPSF and SPSF/GLU membranes were enriched with sulfonate groups or sulfonate and glycosyl groups. The water contact angle decreased from 89.7 ± 3.4° (PSF) to 76.4 ± 3.2° (SPSF) and 71.2 ± 1.9° (SPSF/GLU), respectively. The WCA of SPSF/GLU increased obviously after treatment, indicating that the high hydrophilic dextran was partly dissolved in water, which would prevent the adsorption of LDL from the solution to the membrane surface. BSA protein solution adsorption testing confirmed that the SPSF/GLU membranes with surface enrichment of sulfonate groups and glycosyl groups possessed highest resistance to protein solution among the three studied samples.

The ELISA results indicated that LDL adsorption on PSF was primarily driven by hydrophobic interactions, whereas LDL adsorption on SPSF was mainly induced by electrostatic attraction. SPSF/GLU, which had good anti-nonspecific protein adsorption property, exhibited a similar affinity for LDL compared with PSF. This finding indicated that SPSF/GLU had good affinity for LDL. It could be a potential material for LDL apheresis if the problem of surface dextran loss can be solved. The SPSF film showed an excellent affinity for LDL according to ELISA, along with the simple, time-saving fabrication processes, which made the SPSF membrane a promising economical material for LDL apheresis.

## Figures and Tables

**Figure 1 materials-16-04641-f001:**
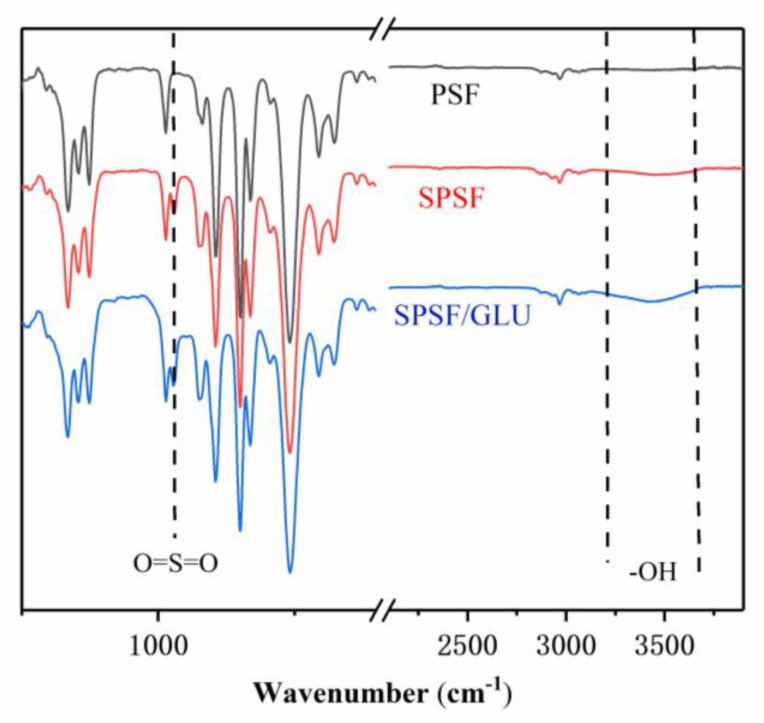
FT-IR spectra of PSF, SPSF, and SPSF/GLU.

**Figure 2 materials-16-04641-f002:**
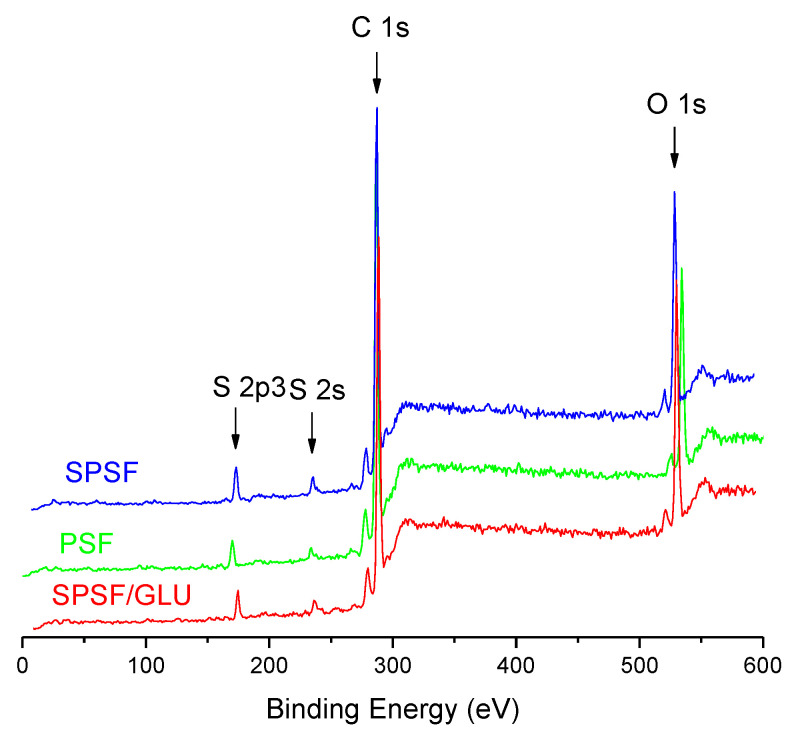
XPS spectra of the PSF, SPSF, and SPSF/GLU surfaces.

**Figure 3 materials-16-04641-f003:**
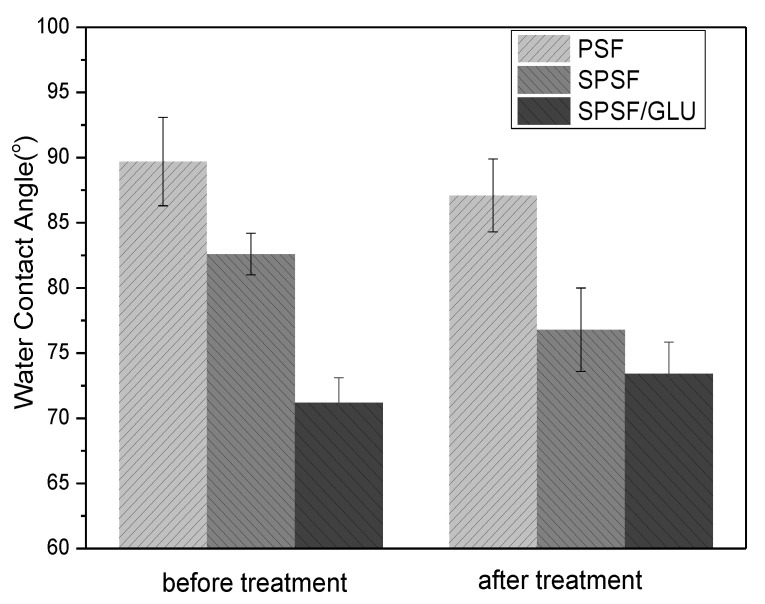
The static water contact angles of PSF, SPSF, and SPSF/GLU before and after treatment (samples were immersed in ultrapure water at 80 °C for 24 h and then dried under a vacuum at 60 °C for 24 h).

**Figure 4 materials-16-04641-f004:**
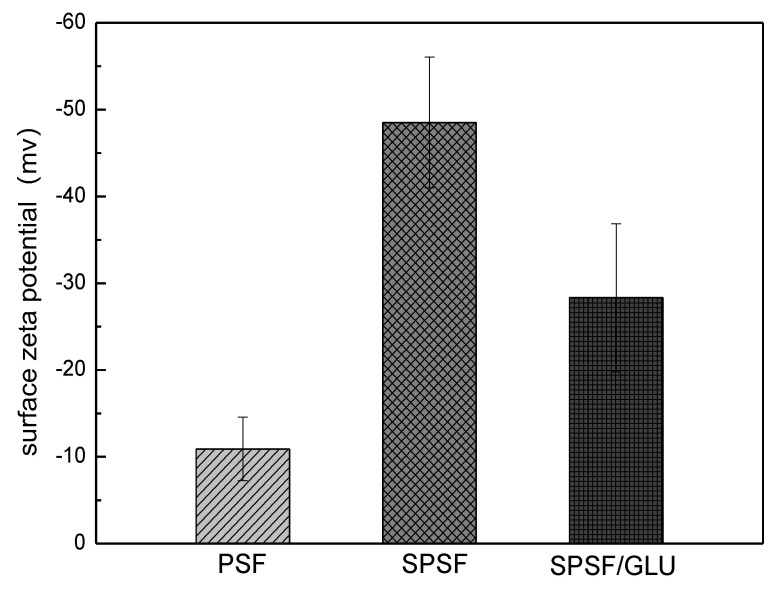
Zeta potential of PSF, SPSF, and SPSF/GLU membrane surfaces.

**Figure 5 materials-16-04641-f005:**
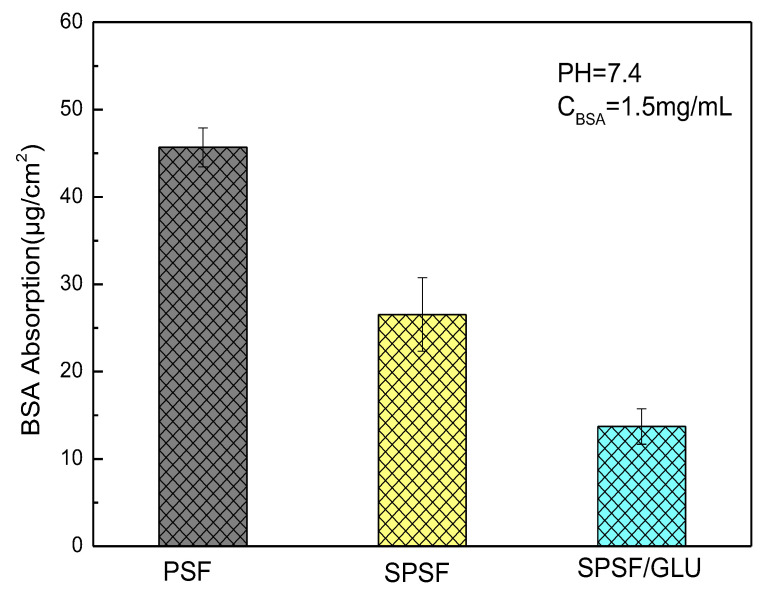
Adsorption of BSA on PSF, SPSF, and SPSF/GLU.

**Figure 6 materials-16-04641-f006:**
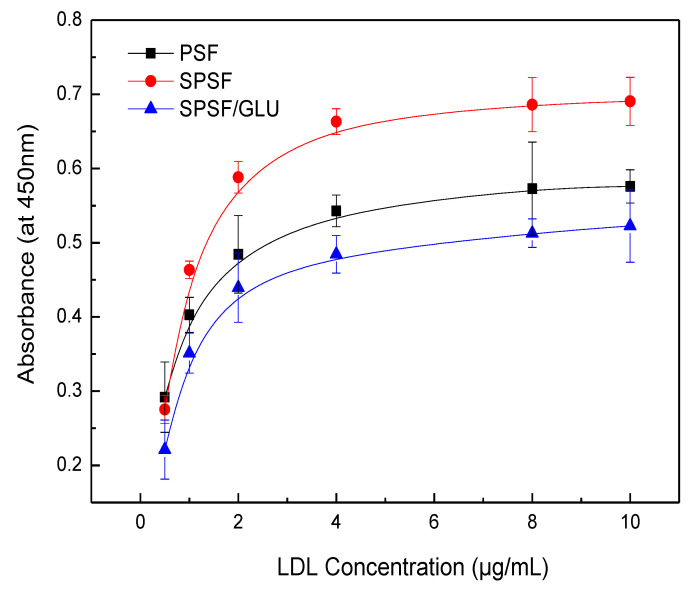
LDL adsorption on PSF, SPSF, and SPSF/GLU in a single protein solution (LDL = 0.5, 1, 2, 4, 8, 10 μg/mL).

**Figure 7 materials-16-04641-f007:**
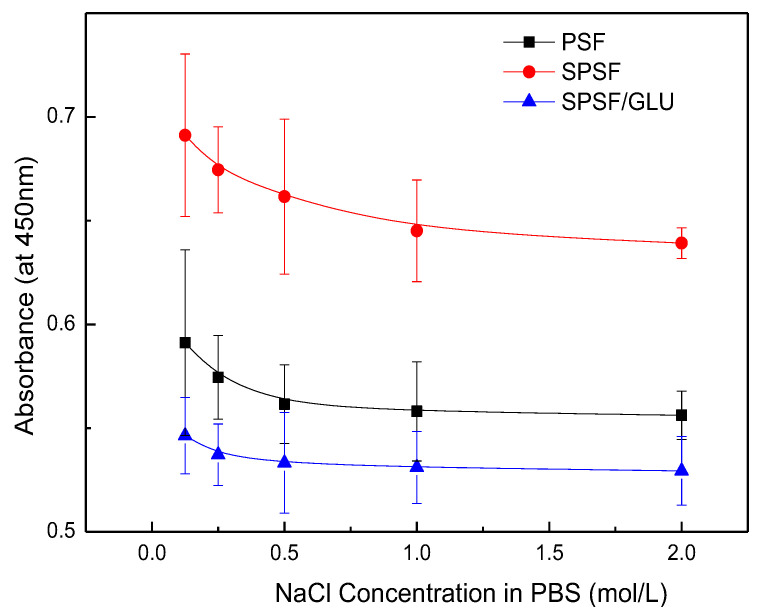
LDL desorption from membranes after exposure to LDL (10 μg/mL) and subsequent washing with NaCl solutions with varying concentrations (NaCl = 0.125, 0.25, 0.5, 1, 2 mol/L).

**Figure 8 materials-16-04641-f008:**
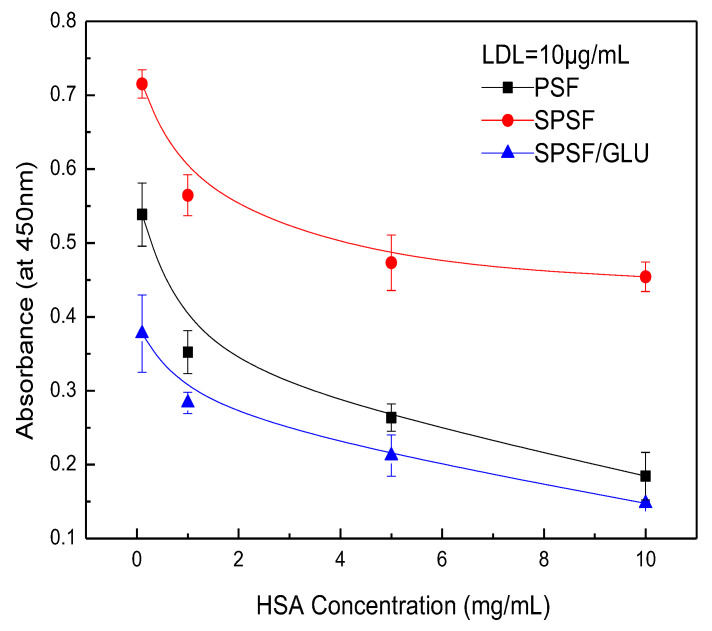
Adsorption of LDL on the PSF, SPSF, and SPSF/GLU from binary protein solutions of LDL and human serum albumin (HSA): LDL = 10 μg/mL, HAS = 0.1, 1.0, 5.0, 10 mg/mL.

**Figure 9 materials-16-04641-f009:**
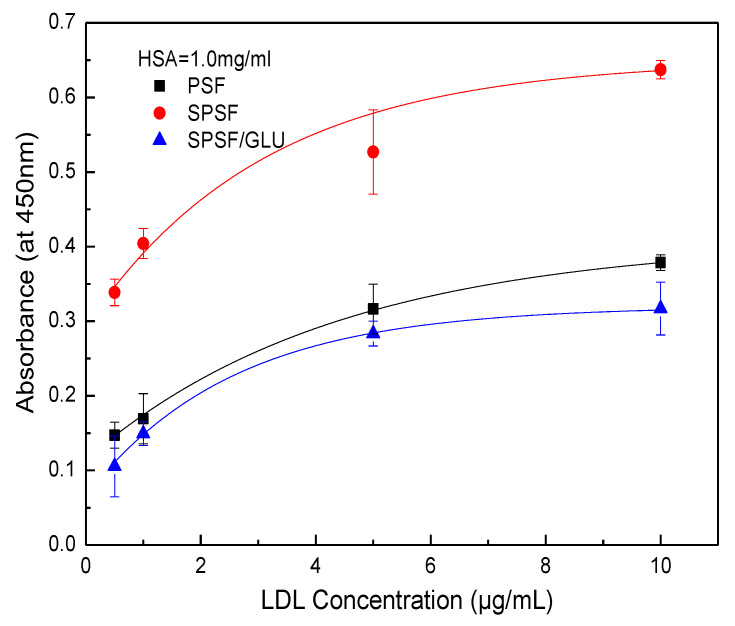
Adsorption of LDL on the PSF, SPSF, and SPSF/GLU from binary protein solutions of LDL and human serum albumin (HSA): HSA = 1.0 mg/mL, HAS = 0.5, 1.0, 5.0, 10 μg/mL.

**Table 1 materials-16-04641-t001:** Chemical compositions of the PSF, SPSF, and SPSF/GLU surfaces from the integration of the C 1s, O 1s, and S 2p3 peaks from XPS spectra in Figure 2.

Samples	Content (wt%)		
	C	O	S
PSF	81.81%	15.91%	2.28%
SPSF	77.04%	19.62%	3.32%
SPSF/GLU	75.53%	21.76%	2.71%

## Data Availability

Not applicable.

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
