# Peer review of "Fabrication and Study of Dextran/Sulfonated Polysulfone Blend Membranes for Low-Density Lipoprotein Adsorption"

_materials, 2023, doi:10.3390/ma16134641_

Round 1

Reviewer 1 Report

Dear Editor

Thank you for invitation to review the manuscript entitled "Fabrication and study of dextran/sulfonated polysulfone blend membranes for low-density lipoprotein adsorption ". In this study, the authors described the fabrication of three types of membranes such as polysulfone (PSF), sulfonated polysulfone (SPSF) and sulfonated polysulfone/dextran (SPSF/GLU) membranes for removal of LDL with high efficiency and selectivity.

It is interested work and significant in the topic of biomedical application. Still there are some major comments to the authors should be addressed as in the following report before we can recommend to accept for publication in Materials.

Best Regards

Comments to the Authors:

1.     Abstract: the first sentence “The abnormal increase of LDL in” full name of LDL should be added.

2.     The second sentence is “In this paper, polysulfone (PSF), sulfonated polysulfone (SPSF) and sulfonated polysulfone/dextran (SPSF/GLU) membranes were successfully fabricated by solution casting technique.” Comment: for what the membrane was fabricated?

3.     Line 68, “we used solution casting technique as a simple and versatile method to prepare anionic or anionic glycosylated PSF membranes.” For what the authors prepare this membrane? More details before you can go for characterization etc.,

4.     Line 89, there are some grammatical errors and typos that should be corrected before publication. (manuscript should be check by a native English-speaking), for example “A Solutions of 16 wt.% SPSF in NMP was prepared by”; line 173, “The XPS spectra of the studied samples are present in Figure.2” line 313, “gradually weakened the with the in-

5.     Line 93, “using a casting knife with a thickness 93 of 150 nm.” Are the authors sure that the thickness of the knife is 150 nm or 150 micro-meter!

6.     Figure 1, I suggest to re-drawing the figure by making a suitable distance between three curves to be more clear for the reader to see the peaks of the functional groups.

7.     Line 128, “was obtained for absorbance at 380 nm (Figure S1).” It is wrong BSA wavelength is 280nm as in Figure S1.

8.     The schematic diagram of the ELISA for LDL adsorption and desorption process should be added even it was presented in refs. 25-27.

9.      Line 202, “further exposure of hy-202 drophilic sulfonic acid groups after treatment.” While in line 218, the authors stated “reduced due to the relatively lower sulfate groups covering the surface layer.

There are some grammatical errors and typos that should be corrected before publication. (manuscript should be check by a native English-speaking),

Author Response

Revision and explanation for the comments of Reviewer 1:

Thank you for invitation to review the manuscript entitled "Fabrication and study of dextran/sulfonated polysulfone blend membranes for low-density lipoprotein adsorption ". In this study, the authors described the fabrication of three types of membranes such as polysulfone (PSF), sulfonated polysulfone (SPSF) and sulfonated polysulfone/dextran (SPSF/GLU) membranes for removal of LDL with high efficiency and selectivity.

It is interested work and significant in the topic of biomedical application. Still there are some major comments to the authors should be addressed as in the following report before we can recommend to accept for publication in Materials.

Comment to the Authors:

  1. Comment: Abstract: the first sentence “The abnormal increase of LDL in” full name of LDL should be added.

Revision and explanation: Thanks for your helpful comment, and we must apologize for this mistake. The full name of LDL “low-density lipoprotein” has been added accordingly in the first sentence in Abstract section in Page 1, 10th line.

  1. Comment: The second sentence is “In this paper, polysulfone (PSF), sulfonated polysulfone (SPSF) and sulfonated polysulfone/dextran (SPSF/GLU) membranes were successfully fabricated by solution casting technique.” Comment: for what the membrane was fabricated?

Revision and explanation: Thanks for this helpful advice.

The second sentence at the abstract section In this paper, polysulfone (PSF), sulfonated polysulfone (SPSF) and sulfonated polysulfone/dextran (SPSF/GLU) membranes were successfully fabricated by solution casting technique. was revised into “In this paper, polysulfone (PSF), sulfonated polysulfone (SPSF) and sulfonated polysulfone/dextran (SPSF/GLU) membranes were successfully fabricated for LDL adsorption by solution casting technique.” from Page 1, 13th line to Page 1, 15th line.

  1. Comment: Line 68, “we used solution casting technique as a simple and versatile method to prepare anionic or anionic glycosylated PSF membranes.” For what the authors prepare this membrane? More details before you can go for characterization etc.,

Revision and explanation: Thank you for this valuable advice. First of all, we are sorry for our puzzling descriptions on the reason why we prepared membranes. To clearly illustrate the issue, additional details have been incorporated in Page 2, 72th line to Page 2, 89th line as follows:

“Gennerally speaking, the selection of ligands and substrates, as well as the preparation methods, are crucial for the development and application of efficient LDL adsorption materials.  Previous studies have shown that anionic or anionic-glycosylated ligands often exhibit excellent LDL adsorption capabilities. Various strategies have been developed to achieve facile and reliable immobilization of anionic or anionic-glycosylated biomaterial surfaces. Despite their good LDL adsorption performance, the preparation process for these adsorbents is often cumbersome, typically involving chemical methods for substrate calcination., which severely limits the practical application of LDL materials. Solution casting is a widely-used technique for producing polymer films, coatings, and membranes. It offers convenience, cost-effectiveness, and time-saving benefits, allowing for the regulation of material properties through the utilization of additives. Moreover, it can be readily adapted for both laboratory-scale research and large-scale industrial production.

     Polysulfone is currently the most widely applied polymer and our research group has devoted significant efforts to using PSF membranes for improved hemodialysis with simultaneous LDL removal. In this study, we prepared anionic or anionic-glycosylated PSF membranes for LDL adsorption. Sulfonated polysulfone and dextran were used as additives, and the membranes were prepared via the solution casting method.”

  1. Comment: Line 89, there are some grammatical errors and typos that should be corrected before publication. (manuscript should be check by a native English-speaking), for example “A Solutions of 16 wt.% SPSF in NMP was prepared by”; line 173, “The XPS spectra of the studied samples are present in Figure.2” line 313, “gradually weakened the with the in-”

Revision and explanation: Thank you for this valuable advice. And the corresponding part are listed as follows:

Page 3, 112th line to Page 3, 113th line: “A Solutions of 16 wt.% SPSF in NMP was prepared by dissolving pure SPSF granules in NMP.” was changed intoA solution of SPSF at a concentration of 16 wt.% in NMP was prepared by dissolving pure SPSF granules in NMP.”

Page 5, 207th line to Page 5, 208th line: “The XPS spectra of the studied samples are present in Figure.2 and the results calculated from the XPS survey scans are listed in Table. 1.” was changed into “Figure 2 displays the XPS spectra of the studied samples, while Table 1 presents the results derived from the XPS survey scans.

Page 11, 360th line to Page 11, 362th line: “Furthermore, the interference effect on the adsorption of LDL by SPSF is gradually weak-ened the with the increase of HAS concentration.” was changed intoFurthermore, the interference effect of HAS concentration on the adsorption of LDL by SPSF gradually diminishes.

We genuinely cherish the chance to publish our work in this journal, and have carefully checked and improved the English writing in the revised manuscript.

  1. Comment: Line 93, “using a casting knife with a thickness 93 of 150 nm.” Are the authors sure that the thickness of the knife is 150 nm or 150 micro-meter!

Revision and explanation: Thanks for your helpful comment and we must apologize for this simple mistake.

Page 3, 116th line to Page 3, 118th line: “After the solution's air bubbles were removed completely, the SPSF/GLU solution was cast onto a clean glass plate using a casting knife with a thickness of 150 nm. was changed into “When all air bubbles were completely removed from the solution, the SPSF/GLU solution was cast onto a clean glass plate using a casting knife with a thickness of 150 μm.

  1. Comment: Figure 1, I suggest to re-drawing the figure by making a suitable distance between three curves to be more clear for the reader to see the peaks of the functional groups.

Revision and explanation: Thanks for your helpful suggestion! Figure 1 has been redrawn and listed as follows to be more clear for the reader to see the peaks of the functional groups.

Figure 1. FT-IR spectra of PSF, SPSF and SPSF/GLU.

  1. Comment: Line 128, “was obtained for absorbance at 380 nm (Figure S1).” It is wrong BSA wavelength is 280nm as in Figure S1.

Revision and explanation: Thanks for this comment! We are very sorry for our incorrect writing and it is rectified to 280 nm in Page 4, 156th line.

  1. Comment: The schematic diagram of the ELISA for LDL adsorption and desorption process should be added even it was presented in refs. 25-27.

Revision and explanation: Thanks for your helpful suggestion! A schematic diagram of the ELISA was added accordingly as Figure S2. And Page 4, 162th line to Page 4, 163th line: “Adsorption of LDL on the different films were investigated by ELISA as described in our previous work [25-27].” was revised into “Adsorption of LDL on the different films were investigated by ELISA (Figure S2) as described in our previous work [25-27].”  

Figure S2. Standard curves of the BSA adsorption.

  1. Comment: Line 202, “further exposure of hy-202 drophilic sulfonic acid groups after treatment.” While in line 218, the authors stated “reduced due to the relatively lower sulfate groups covering the surface layer.”

Revision and explanation: Thanks for your helpful suggestion and we are sorry for our puzzling statement.

Page 6, 242th line to Page 6, 244th line: “It can be seen that the WCA of PSF membrane remained unchanged, while the WCA of SPSF films decreased to 76.4 ± 3.2° due to the further exposure of hydrophilic sulfonic acid groups after treatment.” In this section, the WCA of SPSF was compared before and after treatment. As we can see from Figure 3 that the WCA of SPSF films decreased from 82.5 ±1.3° (before treatment) to 76.4 ± 3.4° (after treatment). It is mainly due to the further exposure of hydrophilic sulfonic acid groups.

While in line 257, “When dextran was introduced to the membrane surface, the surface potential of SPSF/GLU obviously increased, which means the electronegativity of SPSF/GLU is reduced due to the relatively lower sulfate groups covering the surface layer.” In this section, the electronegativity of SPSF/GLU is compared with the electronegativity of SPSF. As we know, the surface of SPSF/GLU was partly covered by GLU molecule, the content of sulfate groups is relatively lower than SPSF.

In order to express more clearly, Page 6, 239th line to Page 6, 242th line: “It can be seen that the WCA of PSF membrane remained unchanged, while the WCA of SPSF films decreased to 76.4 ± 3.2° due to the further exposure of hydrophilic sulfonic acid groups after treatment.” was changed into “It can be seen that the WCA of PSF membrane remained unchanged, while the WCA of SPSF films decreased from 82.5 ± 1.3° to 76.4 ± 3.2° due to the further exposure of hydrophilic sulfonic acid groups after treatment.” Page 7, 257th line to Page 7, 260th line: “When dextran was introduced to the membrane surface, the surface potential of SPSF/GLU obviously increased, which means the electronegativity of SPSF/GLU is reduced due to the relatively lower sulfate groups covering the surface layer. was changed into “When dextran was introduced to the membrane surface, the surface potential of SPSF/GLU obviously increased, which means the electronegativity of SPSF/GLU is reduced due to the relatively lower sulfate groups covering the surface layer compared with that of SPSF.

Reviewer 2 Report

In this study, Fang et al investigated the effect of polysulfone (PSF), sulfonated polysulfone (SPSF) and sulfonated polysulfone/dextran (SPSF/GLU) membranes on low-density lipoprotein adsorption, and found that SPSF/GLU possessed a similar adsorption capacity compared with PSF. I have the following concerns:

1.      As the goal of treating dyslipidemia is to decrease LDL and increase HDL. Therefore, the effect of the membranes on HDL absorption should be investigated as well.

2.      For Figure 6 LDL adsorption on PSF, SPSF and SPSF/GLU in a single protein solution, the absorbance of LDL without the membrane (control) should be added.

3.      Why BSA adsorption experiment was conducted at 30 C but not 37C.

4.      The Discussion section is missing

Author Response

Revision and explanation for the comments of Reviewer 2:

In this study, Fang et al investigated the effect of polysulfone (PSF), sulfonated polysulfone (SPSF) and sulfonated polysulfone/dextran (SPSF/GLU) membranes on low-density lipoprotein adsorption, and found that SPSF/GLU possessed a similar adsorption capacity compared with PSF. I have the following concerns:

Comment to the Authors:

  1. Comment: As the goal of treating dyslipidemia is to decrease LDL and increase HDL. Therefore, the effect of the membranes on HDL absorption should be investigated as well.

Revision and explanation: Thanks for this helpful comment. Firstly, the reviewer's viewpoint is absolutely correct. An ideal LDL-affinity adsorption membrane often aims to adsorb LDL while minimizing the adsorption of beneficial component such as HDL and HSA. The main reasons for not addressing the membrane's adsorption of HDL in this paper are as follows:

This paper is aimed to simplify the preparation of LDL affinity membranes via solution casting technique as a simple and versatile method, thus expanding their practical applications. Additionally, to explores the adsorption effects of different types of membranes on LDL and related factors.

In this paper, the adsorption of LDL is investigated by ELISA and the schematic diagram of the ELISA is shown in Figure S2. The primary antibody is the antibody against LDL, and the second antibody is the antibody against the primary antibody, which is conjugated with peroxidase. Analysis was performed through the color reaction between the enzyme and substrate. For HDL and LDL, the corresponding primary antibody and second antibody are totally different. It would be very difficult to simultaneously determining the amounts of LDL and HDL adsorbed on the membrane from the color reaction.

Furthermore, before conducting the ELISA experiment, we also need to explore the appropriate concentrations and ratios of the primary antibody and the second antibody.  When the primary antibody and the second antibody for different proteins are simultaneously added, the situation can become very complex and difficult to predict.

Figure S2. Standard curves of the BSA adsorption.

Comment: For Figure 6 LDL adsorption on PSF, SPSF and SPSF/GLU in a single protein solution, the absorbance of LDL without the membrane (control) should be added.

Revision and explanation: Thanks for this comment.

First of all, the adsorption data of different membranes have already been corrected by subtracting the background adsorption of the blank plate. Moreover, before the final measurement of LDL on different membrane surfaces, the membranes with adsorbed LDL were transferred to a new 24-well plate as described in the experimental section in Page 4, 180th line “Subsequently, the films were transferred to a new 24-well plate followed by the addition of TMB substrate solution.” The absorbance of LDL without the membrane can be neglected. That’s why the absorbance of LDL without the membrane (control) were not added. Additionally, to prevent direct adsorption of LDL on the well plate surface, a small molecular protein BSA is added as a blocking agent after the incubation of LDL protein.

  1. Comment: Why BSA adsorption experiment was conducted at 30 C but not 37C.

Revision and explanation: Thanks for your helpful suggestion. The BSA adsorption experiment was conducted at 37℃ instead of 30℃. we are very sorry for causing you inconvenience with such a writing error.

Page 4, 156th line to Page 4, 157th line:In a typical experiment, the film was immersed in 4 mL of the BSA solution and was gently shaken for 2 h at 30 °C.”  has been revised to “In a typical experiment, the film was immersed in 4 mL of the BSA solution and was gently shaken for 2 h at 37 °C.”

  1. Comment: The Discussion section is missing

Revision and explanation: Thanks for your advice. Due to our writing error, which led to a misunderstanding on your part, the discussion section has already been included in the third results section. The title of the third section “3.Results” has been changed to. “3. Results and Discussion” in Page 4, 190th line and the title of the fourth section “5. Conclusion” has been changed to. “4.Conclusion” in Page 5, 386th line.

Round 2

Reviewer 1 Report

Dear Editor

I would like to inform you that the authors answer all of the reviewers comments therefore, I recommend to accept it for publication in Materials

Regards

Reviewer 2 Report

Thanks for addressing my concerns.